# Tourism Competitiveness versus Sustainability: Impact on the World Economic Forum Model Using the Rasch Methodology

Vidina Tais Díaz-Padilla *, Irena Travar [ID], Zamira Acosta-Rubio [ID] and Eduardo Parra-López [ID]

Department of Business Management and Economic History, Faculty of Economics, Business and Tourism, Universidad de La Laguna, 38206 Santa Cruz de Tenerife, Spain; alu0101382388@ull.edu.es (I.T.); zacosta@ull.edu.es (Z.A.-R.); eparra@ull.edu.es (E.P.-L.)
* Correspondence: vdiazpad@ull.edu.es

**Abstract:** The pandemic changed the strategic business approach of tourist destinations on a global scale. Given this new scenario, there is a need to implement sustainability strategies that are aligned with economic, social, and environmental aspects to continue competing in the international tourism market. Therefore, identifying these strategies, specific to each destination, is a key variable for tourism competitiveness. To help destination managers, this paper aims to measure tourism competitiveness in terms of sustainability. Using the Rasch model, the analysis confirms that the Sustainable Development Goals (SDGs) represent and, thus, measure tourism competitiveness. In addition, the results obtained show that the countries with the highest socioeconomic development are the most competitive tourist destinations, and the most relevant SDGs for tourism competitiveness are related to prosperity and social guarantees.

**Keywords:** sustainable development; tourist destinations; competitiveness; Rasch model; World Economic Forum

## 1. Introduction

The sustainability of planet Earth has been in danger for several decades, and the awareness of society is increasing, demanding and valuing measures in this regard. In this sense, the tourism sector is no exception. The sustainability of tourism activity has been positioned in recent years as an important variable for attracting tourists [1,2]. In general, it can be considered that a country is competitive when it has the capacity both to increase its levels of productivity and to manage its available resources in such a way that the population can enjoy high levels of well-being and employment without increasing its foreign debt [3]. Certainly, when tourist destinations are countries, tourism competitiveness is directly related to the destination's ability to attract and satisfy tourists [3–9]. Moreover, taking into consideration the importance of the arrival of tourists for tourism competitiveness, the sustainability of the tourist destination is configured as a key competitive variable.

According to the United Nations (UN), sustainability is defined as "meeting the needs of the present without compromising the ability of future generations to meet their Rown needs" [10]. However, the implementation of sustainability principles is a challenge for countries, which is why the UN has established a strategy in this regard. The first step was taken in September 2000 when the Millennium Declaration was signed and the definition of the Millennium Development Goals (MDGs) was established. The results achieved for these MDGs were reviewed in 2012 at the Rio+20 Conference. The outcome was the introduction of a new group of goals, broader both in their definition and in their compliance deadline [11]. Therefore, it was not until September 2015, when, at the UN General Assembly, the 2030 Agenda for Sustainable Development was adopted, defining 17 Sustainable Development Goals (SDGs). These SDGs, made up of 169 targets and universal in nature, have an expected completion date of 2030. They are aimed at ending poverty, fighting for peace and against inequality and injustice, bringing climate change to

an end, and protecting the environment and include the sustainability of cities and urban settlements [11,12].

There have been several approaches to measuring the sustainability of tourism at a destination level. For instance, the Tourism Sustainable Development Index (TSDI) is the only method that provides satellite data for 189 countries and, at the national level, gives the appropriate ranking [13]. On the other side, the UNWTO initiated the formulation of a statistical framework for Measuring the Sustainability of Tourism (MST) in order to determine to what extent tourism contributes to sustainable development both at the national and subnational level [14].

Despite the universal nature of the SDGs, it should be clarified that countries set their own goals according to the general criteria established in the 2030 Agenda [15]. Therefore, a problem arises when defining global indicators that are (i) quantifiable and measurable and, in turn, are (ii) consistent with the 2030 Agenda to reflect the fulfillment of each SDG in each country or region. This paper aims to provide an alternative for the measurement of the SDGs that allows solving these difficulties based on the measurement of tourism competitiveness identified by the World Economic Forum (WEF) through the Travel and Tourism Competitiveness Index (TTCI) [16]. In this context, the model obtained in this study is very useful for tourism destination managers involved in decision-making, whether they are public administrations or private agents. The versatility of its application, as well as the possibility of having official statistical data, position it as a key tool to identify critical areas of action in terms of competitiveness.

## 2. The World Economic Forum and Tourism Competitiveness Index

The global travel and tourism sector has been following a growing trend, which has caused certain risks for local communities and the environment. Thus, the Travel and Tourism Competitiveness Report by the WEF helps tourism stakeholders identify strengths and areas for improvement to establish a sustainable base that will help overcome the negative effects of tourism in the future. Along with this, the needs of tourism demands are covered, as well [17].

There have been different initiatives by the World Travel and Tourism Council (WTTC) to develop a model to measure destination competitiveness. In this context, three editions of the Competitiveness Monitor were published between 2001 and 2004 [18]. The competitiveness index was developed to measure the following indicators: price, infrastructure, environment, technology, human resources, level of openness, social development, and human tourism [19]. Another important document is the Global Competitiveness Report, first launched in 1979 and published annually since then [20]. These two documents represent the methodological foundation for the Travel and Tourism Competitiveness Report, biennially published since 2007 [18].

The result of the extensive research and analysis by the WEF is the TTCI measured for 140 economies/countries. "The index is comprised of four subindexes, 14 pillars and 90 individual indicators, distributed among the different pillars" [17] (p. vii). The four subindexes include enabling environment, T and T policy and enabling conditions, infrastructure, and natural and cultural resources. The dataset is comprised of statistical data from international organizations and survey data from the WEF's annual executive opinion survey, two-thirds and one-third, respectively [17].

A review of the indicators and pillars is necessary [21] to find the most appropriate and reliable ones that will undoubtedly represent the level of destination competitiveness. Since the first Travel and Tourism Competitiveness Report in 2007, the methodological approach has remained almost the same, while the number of indicators has increased. There were 58 indicators in 2007, while eight years later, in 2015, there were 90 indicators. Other changes include a new pillar that was added in 2009, "Affinity for Travel & Tourism", several changes in data sources, and the way the pillars were arranged in sub-indices [22] (p. 5).

The main conclusion of the report published in 2019 is that global connectedness, which is crucial for the development of the travel and tourism sector, is improving. In other words, the indexes that measured air transport infrastructure, digital connectivity, and international openness showed significant increases. However, some concerns arose related to the growing demand for aviation services. Namely, if the growth trend continues, the infrastructure will not be able to respond accordingly due to the lack of capacities [17].

The countries ranking demonstrated that out of 132 economies that were included in the reports in 2017 and 2019, 101 economies had better score results. Moreover, it was proved that the TTCI is strongly related to economic productivity. That means that higher TTCI scores encompass better economic productivity [17].

Regarding sustainability, tourism carrying capacity has come to the fore. In this context, all tourism activities need to be planned through cross-industry and public–private collaboration in a way that does not surpass the destination's carrying capacity. The analysis of indicators that constitute Pillar 9, which refers to environmental sustainability, showed that the number of UNESCO natural and cultural sites increased, as did the number of environmental treaty ratifications. This has resulted in an increase in the index score for the mentioned pillar [17].

Even though the TTCI is seen as "the best existing index in terms of comprehensiveness and methodological development at international level" [23] (p. 254), many scholars have criticized its credibility [21–25]. For instance, Crouch [24] stated that "the WEF index is applicable only at a national level and is limited in the attributes it considers" (p. 4). Kunst and Ivandić [22] proved that TTCI is not a reliable measure because the changes in its score do not imply an appropriate change in tourism activities. Moreover, these authors claimed that another important drawback is proxy variables. In other words, proxy values are used for indicators that cannot be measured. On the other side, Kayar and Kozak [25] indicated that the TTCI does not include all the necessary determinants to measure destination competitiveness. In addition, they do not agree that all the pillars are weighted equally since their contribution to tourism development is different. Kester and Croce [21] concluded that developed countries are ranked higher because they started tourism development earlier and, thus, have more resources and more time to deal with fundamental issues related to tourism development. They suggested that the TTCI index should measure improvement over time, and not only the current situation. Magrini and Grassini [23] highlighted two main limitations: the way a sub-index is computed and a situation where the data are unavailable. Their solution includes the application of partial least squares path models to biannual panel data.

Some authors have outlined the importance of the sustainability concept and agreed that it should be included in all the pillars [26,27]. Rodríguez-Díaz and Pulido-Fernández [27] argued that the WEF ranking is not reliable since some countries are ranked higher due to good scores for certain pillars; however, they may lack sustainable principles. Thus, they proposed a synthetic indicator to measure competitiveness, which takes into consideration the achieved levels of sustainability among pillars. Hassan [26] created a destination competitiveness model that encompasses environmental sustainability factors. More precisely, they identified four determinants of market competitiveness: comparative advantage, demand orientation, industry structure, and environmental commitment.

Furthermore, some authors used models different from the WEF's to measure destination competitiveness [18,28–31]. In this way, Abreu–Novais et al. [28] applied phenomenographic analysis and found three different conceptions of destination competitiveness, which are interconnected and are in a hierarchical relationship: destination competitiveness as perception of a destination, performance, and a long-term process. Croes and Kubickova [29] created the TCI ranking based on satisfaction, productivity, and quality of life. This ranking differs from the one by the WEF. These authors stated that it is a more relevant measure because a higher-ranking position means more tourism receipts per capita and better quality of life, which is not the case with the WEF tourism ranking. Gómez-Vega and Picazo-Tadeo [30] used a weighted composite indicator, where the weights were obtained

endogenously. However, their ranking is similar to the ranking of the WEF. Alves and Nogueira [18] measured sustainable tourism competitiveness (STC) by creating a model that was based on the WEF's model and, by applying partial least squares path modeling (PLS-PM), they showed that tourism infrastructure had the greatest impact on the STC, followed by heritage and culture, ICT infrastructure, and development constructs. Pulido-Fernández and Rodríguez-Díaz [31] used a multi-objective method of double reference points to create an index that helps tourism stakeholders identify specific pillars that need to be improved for each country separately.

## 3. Research Methodology

### 3.1. Relations between the WEF and SDGs

In this paper, the starting point for measuring the accomplishment of the SDGs of the tourist destinations considered is to use the data provided by the WEF to measure tourism competitiveness (Figure 1). In this regard, relationships have been established between the 17 SDGs and those pillars or WEF indicators (90) that could represent the progress of the countries in achieving the SDGs.

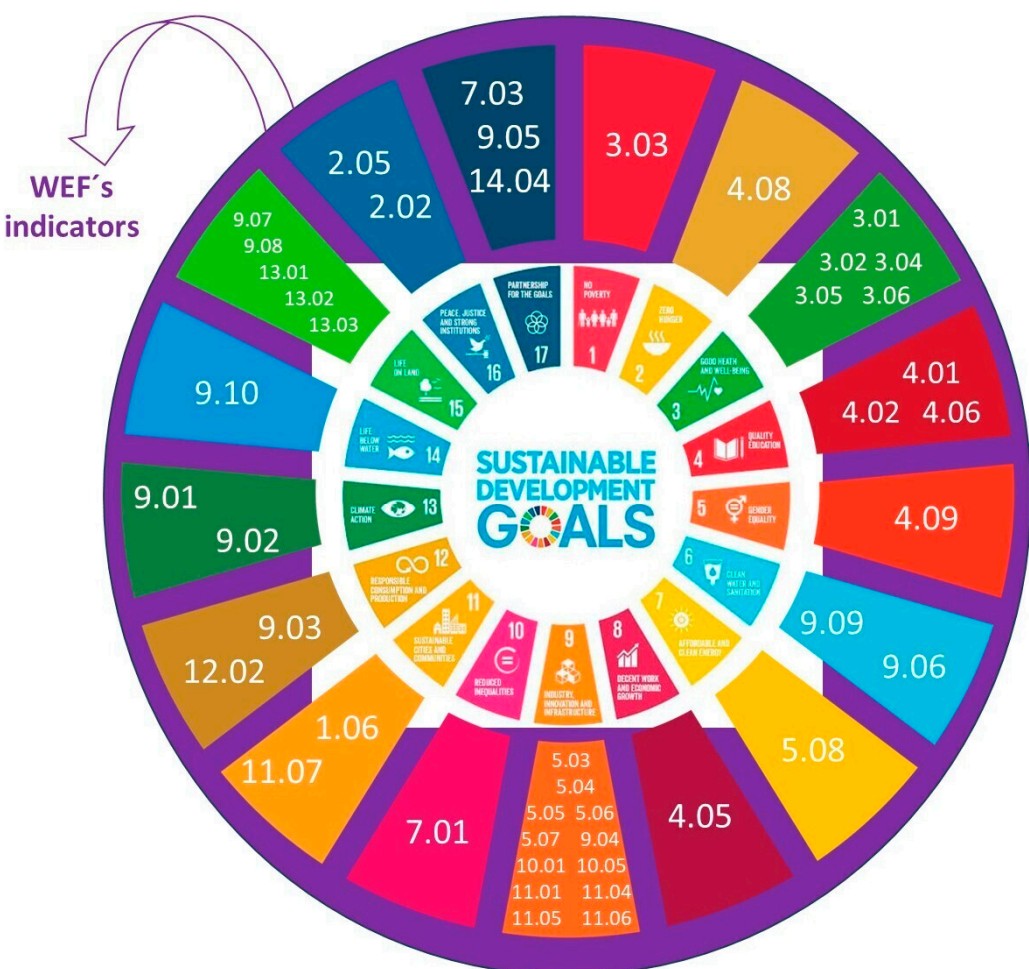

**Figure 1.** Identification of SDGs and WEF indicators [11,15,17].

Related to SDG 1, whose central point is providing essential services to all people in the world and, in this way, eliminating poverty, Indicator 3.03 of the WEF, which measures the use of basic drinking water, can reflect the access to basic services for poor and vulnerable populations.

Regarding SDG 2, focused on supporting agriculture, Indicator 4.08 of the WEF, pay and productivity measurement, offers information on agricultural productivity from which appropriate policies can be established to improve the conditions of small food producers.

SDG 3 ensures healthy lives and promotes well-being for all at all ages. That can be related to the quality of health services or the prevalence of certain diseases, like HIV or malaria. In this case, the WEF proposes five key indicators: 3.01, 3.02, 3.04, 3.05, and 3.06, which measure physician density, use of basic sanitation, hospital beds, HIV prevalence, and malaria incidence, respectively.

SDG 4, establishing quality education systems, can be measured through Indicators 4.01, 4.02, and 4.03 of the WEF. These indicators measure the primary education enrolment rate, secondary education enrolment rate, and the ease of finding skilled employees, respectively.

With reference to SDG 5, assessing female labor force participation using Indicator 4.09 of the WEF can be useful in reducing gender inequality and eradicating discrimination against women and girls everywhere.

In relation to SDG 6, concerning available and quality water and sanitation everywhere in the world, baseline water stress and wastewater treatment can be used for its measurement. These are Indicators 9.06 and 9.09 of the WEF, respectively.

SDG 7, related to the widespread use of sustainable energy sources, can be related to Indicator 5.08 of the WEF, which measures the quality of electricity supply.

In connection with SDG 8, Indicator 4.05 of the WEF, hiring and firing practices, can be used to measure whether countries fulfill the necessary conditions for providing decent working environments and ensuring labor rights. This is one of the preconditions for the accomplishment of long-term sustainable economic growth.

About SDG 9, there are numerous WEF indicators to assess compliance with the sustainable infrastructure in the countries considered. In this sense, the large number of WEF indicators related to the development of reliable, sustainable, resilient, and quality infrastructures, including regional and cross-border infrastructures, stands out. Among them are Indicator 10.01, which measures the quality of air transport infrastructure; Indicator 10.05, which indicates the density of airports per million inhabitants; and Indicator 11.01, which refers to the quality of roads. The quality of the railroad infrastructure is also considered in Indicator 11.04, as well as the density of railroad infrastructure through Indicator 11.05 and the quality of port infrastructure with Pillar 11.06. In relation to the adoption of technological innovations and the use of green technologies in industries, among the WEF indicators, there is a reference to the measurement of emissions in Indicator 9.04, which measures particulate matter (2.5) concentrations. Finally, reference should be made to a large amount of data regarding access to information and communications technology, as well as striving to provide universal and affordable access to the internet. Indicators 5.03, 5.04, 5.05, 5.06, and 5.07 of the WEF refer, respectively, to individuals using the internet, broadband internet subscriptions, and mobile telephone and broadband subscriptions, as well as the percentage of the population with mobile network coverage.

In reference to SDG 10, aimed at aiding the less-developed countries to reduce existing differences in terms of economic development, there are data related to facilitating migration of people through the appropriate measures, like visa requirements, which are contained in Indicator 7.01 of the WEF.

Regarding SDG 11, focused on the transformation of human settlements to make them more sustainable, particularly in terms of the modification of transport systems, the WEF provides relevant data regarding ground transport efficiency in Indicator 11.07. In addition, in reference to the need to provide support so that sustainable and resilient buildings can be built, Indicator 1.06 can be observed, referring to the cost of obtaining construction permits.

In relation to SDG 12, aimed at guaranteeing sustainable consumption and production patterns through various channels, among which are specific tools for monitoring sustainable tourism development, two WEF indicators can be extracted. The first one is 9.03,

which allows measuring the sustainability of travel and tourism industry development, and the second one is 12.02, which indicates the quality of tourism infrastructure.

SDG 13, which focuses on fighting climate change and its consequences by acting, is also represented by WEF indicators. These are 9.02 and 9.01. Respectively, these indicators report on the enforcement and stringency of environmental regulations.

Next, SDG 14 is related to the sustainable maintenance of the seas and oceans, as well as their resources and marine life. To evaluate it, WEF Indicator 9.10 can be considered, which refers to fishing pressure on the coastal shelf and, thus, examines which countries are managing their fishing resources more efficiently.

SDG 15 is oriented to terrestrial ecosystems, establishing actions for their conservation that allow stopping the advance of desertification, the destruction of forests, and maintaining biodiversity. For its analysis, there are also a variety of WEF indicators. Specifically, Indicator 9.08 indicates forest cover change, while 13.01 and 13.03 make it possible to verify, respectively, how many natural World Heritage Sites and protected areas there are in the analyzed countries. Likewise, it is also possible to have data on known species and threatened species based on Indicators 13.02 and 9.07, respectively.

To assess compliance with SDG 16, which promotes inclusiveness in societies with widespread justice in peaceful environments with responsible and committed institutions, Indicators 2.05 and 2.02 could be used, which correspond to the ratio of homicides per 100,000 inhabitants and the reliability of police services, respectively.

Finally, SDG 17 favors alliances between countries to achieve more sustainable development. For its evaluation, it is considered that Indicators 7.03, 9.05, and 14.04 of the WEF are useful. They refer to how many regional trade agreements are in force, as well as environmental treaty ratifications and international association meetings, respectively.

### 3.2. Objectives and Methodology

The main objective of this research is to measure tourism competitiveness, which is determined by the level of sustainability in each tourist destination. It should be clarified that, in this paper, tourist destinations refer to both countries and national economies. Next, the specific objectives that arise are:

- Confirming that the SDGs are determinants of tourism competitiveness;
- Identifying the SDGs that contribute to a higher level of tourism competitiveness;
- Determining the most competitive tourist destinations in terms of sustainability.

To achieve the proposed objectives, the evaluation of compliance with the SDGs has been proposed based on a series of indicators defined by the WEF to calculate the TTCI, which are shown in Table 1.

These indicators were used to calculate the most recent TTCI available to date, which corresponds to the year 2019. The sample of countries to be analyzed was obtained from the WEF website (www.weforum.org accessed on 26 May 2021), where a sample of 140 countries is considered (N = 140).

In this paper, the SDGs have been named according to the numbering established by the UN. However, to reduce the confusion that may arise with the results provided by the software used (Winsteps) [32], the SDGs have been coded, as shown in Table 2.

For its part, the WEF considers 90 indicators, of which a total of 45 were used in this research (see Table 1) to represent the SDGs. The data from these indicators have been normalized on a Likert scale (1-7) following the methodology applied by the WEF [17] (p. 86). It is important to emphasize that the WEF normalizes those values whose high level represents a worse outcome for tourism competitiveness (e.g., HIV prevalence).

**Table 1.** TTCI structure.

| The Travel and Tourism Competitiveness Index (TTCI): 4 Sub-Indexes and 14 Pillars | | | |
|---|---|---|---|
| **Enabling Environment** | **T and T Policy and Enabling Conditions** | **Infrastructure** | **Natural and Cultural Resources** |
| Pillar 1: Business Environment— 12 indicators | Pillar 6: Prioritization of Travel and Tourism— 6 indicators | Pillar 10: Air Transport Infrastructure— 6 indicators | Pillar 13: Natural resources— 5 indicators |
| Pillar 2: Safety and Security— 5 indicators | Pillar 7: International Openness— 3 indicators | Pillar 11: Ground and Port Infrastructure— 7 indicators | Pillar 14: Cultural Resources and Business Travel— 5 indicators |
| Pillar 3: Health and Hygiene— 6 indicators | Pillar 8: Price Competitiveness— 4 indicators | Pillar 12: Tourist Service Infrastructure— 4 indicators | |
| Pillar 4: Human Resources and Labor Market— 9 indicators | Pillar 9: Environmental Sustainability— 10 indicators | | |
| Pillar 5: ICT Readiness— 8 indicators | | | |

**Table 2.** SDGs coding.

| SDGs | | Code |
|---|---|---|
| **Number** | **Name** | |
| 1 | No Poverty | A |
| 2 | Zero hunger | B |
| 3 | Good health and well-being | C |
| 4 | Quality education | D |
| 5 | Gender equality | E |
| 6 | Clean water and sanitation | F |
| 7 | Affordable and clean energy | G |
| 8 | Decent work and economic growth | H |
| 9 | Industry, innovation, and infrastructure | I |
| 10 | Reduced inequalities | J |
| 11 | Sustainable cities and communities | K |
| 12 | Responsible consumption and production | L |
| 13 | Climate action | M |
| 14 | Life below water | N |
| 15 | Life on land | O |
| 16 | Peace, justice, and strong institutions | P |
| 17 | Partnerships for the goals | Q |

Regarding the technique used for the analysis, it is essential to highlight the difficulty in measuring in social sciences. Based on this premise, to achieve the stated objectives, the chosen technique must facilitate the study of the latent variable, defined as tourism competitiveness in terms of sustainability. Therefore, in this work, the Rasch model [33,34] has been used. This technique has been successfully applied in other tourism studies, considering Island tourism destinations [35,36] as well as national tourism destinations [37,38]. This model, which is part of Item Response Theory, proposes that the latent variable is defined by the interaction of the research units (countries) and the items of the measurement instrument (SDGs) [39,40]. As a result of the application of this model, the original data are transformed, going from an ordinal scale to an interval scale, which is expressed in logit units [41].

The logit scale consists of the logarithmic transformation of the probability of a correct answer and can vary from minus to plus infinity. In this way, the data obtained (observed) can be represented in the same linear continuum or dimension [42], with "$\beta$" being the

ability of the countries to comply with the SDGs and "$\delta$" the level of difficulty that each SDG shows to be achieved. Even though it was originally proposed for binary item responses, in this research, the Rasch model [33,34] has been applied to polytomous items (more than two possible alternatives). In this case, the probability that item $i$ and subject $v$ are in a single dimension is [43]:

$$P[Xni = x] = \frac{1}{\gamma} e^{[x(\beta_n - \delta_i) - \sum_{k=1}^{x} \tau_{ki}]}$$

where

$v$ = interviewed country;

$i$ = answered item;

$k$ = category in item $i$ assumed by the respondent;

$\gamma$ = sum of all possible numerators, which arise according to the number of categories of the items;

$x$ = latent variable that represents the response of a respondent to an item of the measurement instrument;

exp = base of natural logarithm;

$\beta$ = parameter of the latent trait of the interviewed countries;

$\delta$ = items parameter;

$\tau$ = threshold parameter, indicates the transition point between two adjacent response categories, that is, the cutoff points of the characteristic curves corresponding to the different response categories in the items [44].

The Rasch model [33,34] has some features that are important to highlight. Firstly, in the Rasch methodology, the data fit the model and not the opposite. The starting point is to propose an ideal mathematical model from the original data from which observed measurements are obtained. This aspect, unlike other methodologies, allows researchers to identify those countries and those SDGs that present a different behavior than expected based on the level of mismatch of the measurements obtained [32]. Secondly, this methodology presents specific objectivity. Namely, the differences between two countries must be independent of the specific SDGs with which they have been measured, and, in turn, the differences between two SDGs must be independent of the individual countries that have valued them [45]. Thirdly, the application of the Rasch methodology does not imply the need for the data sample to follow a normal distribution [16]. In addition, the robustness of the Rasch methodology for small samples has been verified, as well as the statistical quality of the reliability and validity that the methodology itself performs [43,46,47]. Additionally, the Rasch model allows for comparative or benchmarking analyses [4]. This is especially indicated for the achievement of the last two specific objectives set out in this research since benchmarking is a procedure for comparing tourist destinations to identify key areas of action in terms of competitiveness [48]. Finally, the data have been processed using Winsteps 3.92.1 software [32].

## 4. Results

To achieve the first specific objective set out in this research and to determine the statistical significance of the SDGs measured with the WEF indicators, some basic analyses have been carried out in the context of the Rasch model. The obtained results are shown below.

### 4.1. Fit (Validity) of the Data

In the Rasch model [33,34], the data must fit the model and not the other way around. Therefore, it is essential to analyze the validity or degree of adjustment of the data used through fit statistics [49]. The fit statistic is the mean of the residuals, that is, the mean of the difference between the real and the estimated scores by the Rasch model [33,34]. These statistics, which are internal (INFIT) and external (OUTFIT), are expressed as a function of the non-standardized mean square (MSNQ) and the normalized variances of

the residuals (ZSTD) [50,51]. The INFIT statistic collects the internal adjustment of the real scores to the model and is sensitive to the unexpected behaviors presented by the scores of those indicators whose difficulty is close to the competitive ability of the country. From an external point of view, the OUTFIT statistic makes it possible to detect unexpected behavior in the scores of those indicators whose difficulty is far from the competitive ability of the country [52,53].

In this sense, the validity of data will be accepted if the values of the INFIT and OUTFIT statistics are in the interval (0.5, 1.5) ($p < 0.05$) for the MSNQ means, with 1 being the expected mean value and in the interval ($-2$, 2) for the ZSTD normalized variances [54]. In this way, and marked in red, Table 3 shows the INFIT and OUTFIT results obtained for the countries (1.01, 1.03), and Table 4 shows those that present the SDGs (1.05, 1.03). As can be seen, the MSNQ means are very close to one (1.00), which shows the existence of fit or validity. Moreover, it is confirmed by the results obtained in the ZSTD standardized variances ($-0.2$, $-0.1$ for countries; $-0.8$, $-0.9$ for SDGs).

**Table 3.** Countries' summary statistics.

| | Total Score | Count | Measure | Model S.E. | INFIT | | OUTFIT | |
|---|---|---|---|---|---|---|---|---|
| | | | | | MNSQ | ZSTD | MNSQ | ZSTD |
| MEAN | 65.1 | 17.0 | −0.36 | 0.26 | 1.01 | −0.20 | 1.03 | −0.10 |
| MAX. | 90.0 | 17.0 | 1.41 | 0.29 | 3.11 | 4.30 | 5.23 | 6.80 |
| MIN. | 36.0 | 17.0 | −2.33 | 0.25 | 0.21 | −3.60 | 0.23 | −3.40 |
| REAL RMSE = 0.29 | | TRUE SD = 0.72 | | SEPARATION = 2.52 | | PERSON RELIABILITY = 0.86 | | |
| MODEL RMSE = 0.26 | | TRUE SD = 0.73 | | SEPARATION = 2.82 | | PERSON RELIABILITY = 0.89 | | |
| PERSON RAW SCORE-TO-MEASURE CORRELATION = 1.00 | | | | | | | | |
| CRONBACH ALPHA (KR-20) PERSON RAW SCORE "TEST" RELIABILITY = 0.88 − SEM = 3.99 | | | | | | | | |

**Table 4.** SDGs' summary statistics.

| | Total Score | Count | Measure | Model S.E. | INFIT | | OUTFIT | |
|---|---|---|---|---|---|---|---|---|
| | | | | | MNSQ | ZSTD | MNSQ | ZSTD |
| MEAN | 535.7 | 140.0 | 0.00 | 0.09 | 1.05 | −0.8 | 1.03 | −0.90 |
| MAX. | 746.0 | 140.0 | 2.27 | 0.11 | 2.46 | 8.40 | 2.52 | 8.20 |
| MIN. | 263.0 | 140.0 | −1.80 | 0.09 | 0.19 | −9.90 | 1.19 | −9.90 |
| REAL RMSE = 0.11 | | TRUE SD = 0.91 | | SEPARATION = 8.62 | | ITEM RELIABILITY = 0.99 | | |
| MODEL RMSE = 0.09 | | TRUE SD = 0.91 | | SEPARATION = 10.01 | | ITEM RELIABILITY = 0.89 | | |
| PILLAR RAW SCORE-TO-MEASURE CORRELATION = −1.00 | | | | | | | | |

Therefore, the data used in this research, both for countries and for SDGs, fit the Rasch model [16]. Likewise, and based on the measures generated by this model, conclusions with global validity can be drawn.

### 4.2. Reliability

The analysis of the reproducibility or reliability of the data is carried out through Cronbach's alpha in combination with the specific indicators of reproducibility (reliability). The reliability levels vary between 0.00 and 1.00. The closer the result obtained to one, the greater the precision of the measurement [55].

Based on Tables 3 and 4, the values obtained, and indicated in green, in the "reliability" indicator are greater than 85% for the countries and 90% for the SDGs. This means that the level of reliability of the measurements obtained by the model is high. In addition, this high reliability allows us to ensure that the measurements are reliable, consistent, reproducible, and precise [56].

### 4.3. Dimensionality

One of the fundamental characteristics of the Rasch model is assuming that the SDGs only measure tourism competitiveness in terms of sustainability, a latent variable or construct of this research [57]. The presence of more than one dimension could be due to disturbances in the relationship between the indicators or to the existence of indicators that represent other variables, different from tourism competitiveness in terms of sustainability [52,58]. To verify the level of unidimensionality quickly, the ID indicator [59] has been developed in the context of the Rasch model: Person separation real reliability/Person separation model reliability. ID values greater than 0.90 would indicate the existence of a single dimension. In this paper, the result obtained in the ID indicator was 0.97 (ID = 0.97), which indicates a high degree of unidimensionality of the "tourism competitiveness in terms of sustainability" construct.

To further confirm this result, the Winsteps program (3.92.1) [32] allows a more detailed analysis of unidimensionality through the Principal Component Analysis of the Residuals (PCAR) of the SDGs [54]. This analysis allows detecting the existence of other dimensions when the "Rasch factor" has already passed [33,34]. To carry out this analysis, Linacre [60] proposed a "golden rule": (i) the percentage of variance not explained in the first test must be lower than the percentage of variance explained by the indicators; (ii) the variance of the items must be equal to or greater than 4 times the unexplained variance in the first contrast; (iii) the unexplained variance in the first test must be less than 3 (in eigenvalue) and less than 5%; and (iv) the variance in the measurements must be greater than 50% [43].

According to the data obtained and presented in Table 5, three of the five specified premises are fulfilled: (i), (ii), and (iv). Therefore, the unidimensionality of the construct "tourism competitiveness in terms of sustainability" has been confirmed.

**Table 5.** Dimensionality analysis (N = 140).

| Content | Eigenvalue | Observed |
|---|---|---|
| Total raw variance in observations | 38.28 | 100% |
| Raw variance explained by measures | 21.28 | 55.6% |
| Raw variance explained by items | 15.36 | 40.1% |
| Unexplained variance in 1st contrast | 3.34 | 8.7% |

### 4.4. Analysis of the Categories

The application of the Rasch model [33,34] provides, as a result, measurements observed on a logit scale. These observed measures, which can be expressed as interval measures, are classified into categories that coincide with the intervals of the Likert scale (1–7). Therefore, to determine the statistical significance of the model by which the SDGs are represented through the WEF indicators, Linacre [60] proposed a series of assumptions. First, the correlation between the SDGs and the latent variable must be positive and greater than 30%. As can be seen in Table 6, marked in red, there are four SDGs that do not meet this requirement: E (Gender equality), J (Reduced inequalities), N (Life below water), and O (Life on land). In the case of the first three SDGs (E, J, N), a correlation of less than 30% with tourism competitiveness in terms of sustainability could be because they are measured by a single WEF item or indicator. As for SDG O (Life on land), it is measured by five WEF indicators, among which Indicator 9.07 (Threatened species) stands out due to its difficult measurement and interpretation. Removing this indicator from the measurement could correct this low correlation.

**Table 6.** Analysis of the results of the SDGs.

| SDGs | Total Score | Measure (Logits) | Correlation with Logits Values | Importance for Tourism Competitiveness in Terms of Sustainability |
|------|-------------|------------------|--------------------------------|------------------------------------------------------------------|
| N | 263 | 2.27 | 0.29 | |
| Q | 400 | 1.04 | 0.74 | Low |
| F | 449 | 0.66 | 0.56 | |
| I | 450 | 0.65 | 0.89 | |
| O | 452 | 0.64 | 0.18 | |
| B | 491 | 0.35 | 0.75 | |
| M | 502 | 0.27 | 0.82 | Medium |
| H | 513 | 0.18 | 0.36 | |
| C | 539 | −0.01 | 0.66 | |
| L | 576 | −0.30 | 0.72 | |
| J | 587 | −0.38 | 0.25 | |
| G | 595 | −0.44 | 0.84 | High |
| K | 605 | −0.52 | 0.80 | |
| E | 622 | −0.66 | 0.18 | |
| D | 631 | −0.74 | 0.77 | Very high |
| P | 686 | −1.21 | 0.61 | |
| A | 746 | −1.80 | 0.71 | |
| *Mean* | *535.70* | *0.00* | ------ | |

In relation to the second specific objective, which is to identify the SDGs that contribute to a higher level of competitiveness, in Table 6, the SDGs are ordered from least to greatest importance for tourism competitiveness in terms of sustainability. In this way, SDG N (Life below water) is the worst valued (263) while SDG A (No poverty) is the one that is valued the most (746) for the set of countries in the sample. In other words, SDG A (No poverty) and SDG N (Life below water) are the SDGs with the greatest and least importance, respectively, for tourism competitiveness in terms of sustainability.

Following the assumptions of Linacre [60], referring now to the categories of intervals of the applied Likert scale, each category must include at least 10 observations [Freq. (count) ≥ 10]. The third premise proposed by Linacre [60] refers to the regular distribution of the observations [Freq. (%)] and indicates the excellence of the category for performing calibrations.

Based on the results shown in Table 7, the previous premises are strictly fulfilled since the low values presented by categories 1 and 7 are not considered misaligned as they are the extreme categories. Fourth, Linacre [60] points out that the mean measures generated by the Rasch model [33,34] must grow regularly to be useful so that the highest scores generate the highest measures. As can be seen in the "OBSVD Average" column of Table 7, the measurements grow regularly. Fifth, OUTFIT values greater than 2 indicate that the SDGs observations provide the Rasch model with more noise than information, which does not occur in view of the results of the "OUTFIT MSNQ" column in Table 7.

**Table 7.** Analysis of the categories.

| Categ. | Freq. (Count.) | Freq. (%) | Obsvd. Average | OUTFIT MNSQ | Andrich Threshold | Category Measure | Coherence M→C | Coherence C→M |
|--------|----------------|-----------|----------------|-------------|-------------------|------------------|-----|-----|
| 1 | 145 | 6 | −2.08 | 1.49 | NONE | −2.08 | 80% | 23% |
| 2 | 291 | 12 | −1.41 | 1.13 | −2.53 | −1.41 | 40% | 31% |
| 3 | 523 | 22 | −0.94 | 0.85 | −1.74 | −0.94 | 45% | 54% |
| 4 | 611 | 26 | −0.34 | 0.77 | −0.72 | −0.34 | 45% | 60% |
| 5 | 526 | 22 | 0.41 | 0.85 | 0.19 | 0.41 | 47% | 50% |
| 6 | 251 | 11 | 1.08 | 1.00 | 1.42 | 1.08 | 61% | 28% |
| 7 | 33 | 1 | 1.78 | 1.27 | 3.37 | 1.78 | 0% | 0% |

According to Linacre [60], the growth of the measures between categories must be ordered and ascending. In this sense, the "Andrich Threshold" indicator measures the step calibration and should increase regularly to a higher category, as shown in Table 7. Finally, regarding the coherence in the distribution of the categories, the measures must imply the categories (M→C), and the categories must imply the measures (C→M). In this way, according to Linacre [60], the parameter "M→C" indicates the expected percentage of measurements generated by the observations in a category, while the parameter "C→M" indicates the percentage of observations of a category that have been generated by measures of the same category. These parameters must be, and, in this case, they are, greater than 40% in the non-extreme categories, which shows that there is sufficient coherence between the scale and the analyzed sample.

In view of the results obtained, it can be ensured that the first objective has been achieved. The model obtained is valid and reliable, and the SDGs that define it measure, exclusively, the construct "tourism competitiveness in terms of sustainability".

*4.5. Global Competitive Position (Countries)*

To achieve the other two proposed objectives, the Rasch model [33,34] provides a series of tools through the Winsteps program (3.92.1) [32]. These are the Positioning Map or Wright Map and, in a complementary way, the Guttman Scalogram [61], as well as the already mentioned information contained in Table 6, where the analysis of the results of the SDGs is presented. The analysis of the tourism competitiveness of the countries in terms of sustainability is carried out through the Positioning Map or Wright Map, which, in turn, also allows us to identify the SDGs that contribute to a higher level of tourism competitiveness.

The specific analysis of the variables that define the latent variable "tourism competitiveness in terms of sustainability" is made up of several parts. Generally, the application of the Rasch model provides an integrated model in which, graphically, the countries and the SDGs appear positioned according to the measurements obtained (in logits). In this graph, called the Wright Map (Figure 2), the countries are ordered in decreasing order of tourism competitiveness, while the SDGs are ordered from less to greater importance for tourism competitiveness.

According to the information in Figure 2 and the data provided by the Winsteps program (3.92.1) [32], the countries with the highest level of tourism competitiveness in terms of sustainability are Germany (DEU), the United States (USA), Sweden (SWE), The Netherlands (NLD), the United Kingdom (GBR), Finland (FIN), Denmark (DNK), Switzerland (CHE), and Iceland (ISL). As for the least competitive countries in terms of sustainability, Nigeria (NGA), Uganda (UGA), Zimbabwe (ZWE), Mozambique (MOZ), Sierra Leone (SLE), Chad (TCD), and Angola (AGO) stand out. Haiti (HTI) is the country with the lowest tourism competitiveness in terms of sustainability, preceded by Yemen (YEM) and Mauritania (MRT).

In relation to the SDGs that contribute the most to tourism competitiveness in terms of sustainability, SDG A (No poverty) stands out, followed by SDG P (Peace, justice, and strong institutions), with the contribution of SDG D (Quality education) being also noteworthy, as well as the SDG E (Gender equality) and SDG K (Sustainable cities and communities). Among the SDGs that contribute the least to tourism competitiveness in terms of sustainability, SDGs N and Q stand out, which refer to Life below water and the Partnerships for the goals, respectively.

Within the specific analysis, the next step consists of jointly analyzing the previous results with those obtained in the Guttman Scalogram (Figure 3). In this graph, in which the rows correspond to the countries and the columns correspond to the SDGs, the category to which each country's assessment of each SDG belongs is shown.

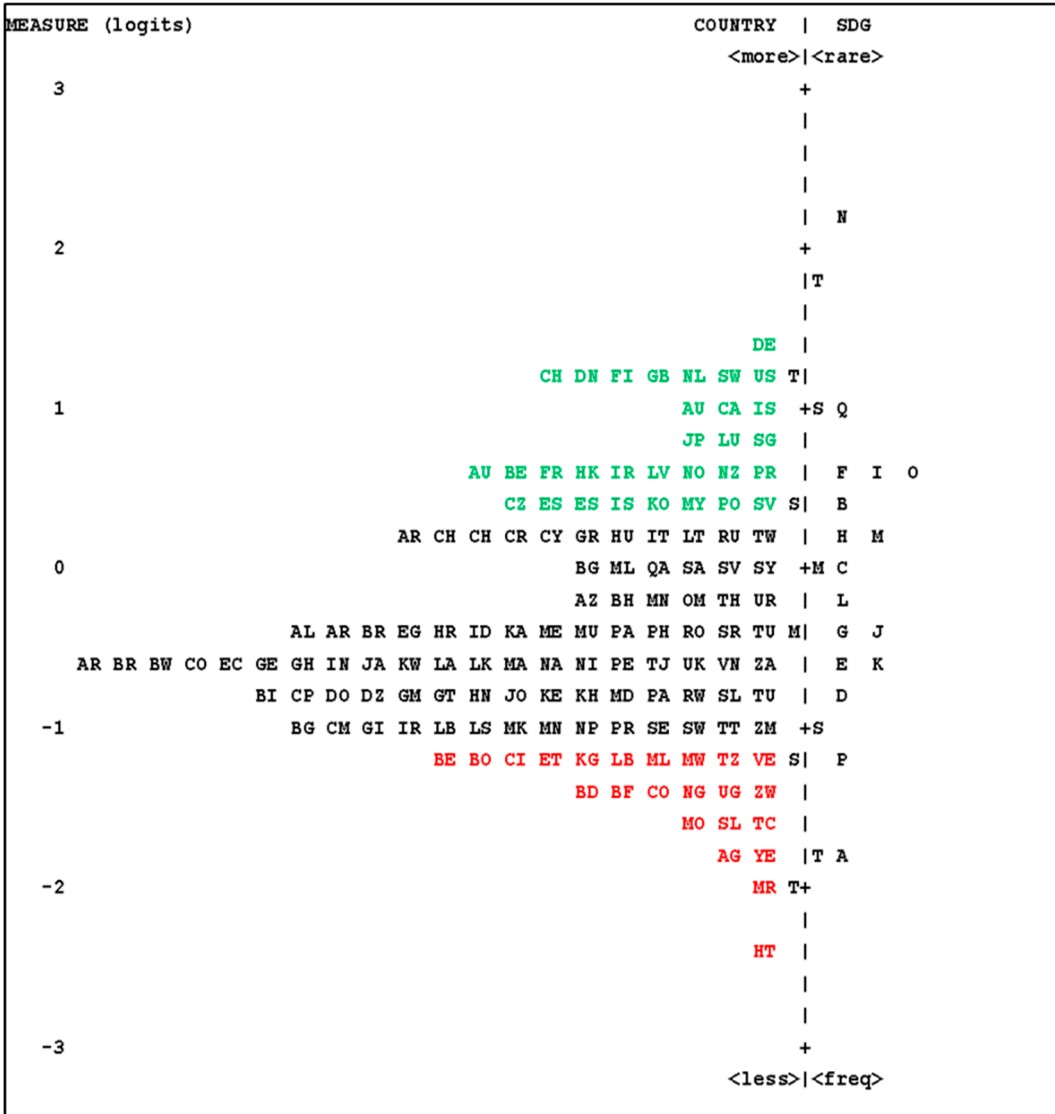

**Figure 2.** The Wright Map.

The response patterns are common among the countries that make up the group of countries with the highest level of socioeconomic development (top) and among the countries that make up the group of countries with the lowest level of socioeconomic development (bottom). However, there are certain SDGs in the countries of both groups that show similar response patterns. This can be seen in Figure 3 and highlighted in red where, based on the assessments, SDG P (Peace, justice, and strong institutions) and SDG E (Gender equality) are achieved by most of the countries in the sample. On the other side are SDG O (Life on land) and SDG N (Life below water) since obtaining low valuations shows the difficulty of achieving them by the countries of both groups.

```
                       Country |SDG
                               |
                               |APDEKGJLCHMBOIFQN
                               |----------------
             Germany (DEU)     +76556555556645564
             Switzerland (CHE) +76556656566635641
             United States (USA) +66656666466744352
             Denmark (DNK)     +76655655456535653
             Finland (FIN)     +76666656537535551
             Netherlands (NLD) +76656655555535553
             Sweden (SWE)      +76665656446535553
             United Kingdom (GBR) +76655655465545562
             Austria (AUT)     +76556656536544651
             Iceland (ISL)     +76655655465534544
             Canada (CAN)      +66555666456544533
             Japan (JPN)       +66556655536435445
                   ...              ...
                   ...              ...
             Burkina Faso (BFA) +25454154233222321
             Burundi (BDI)     +24463262332231421
             Uganda (UGA)      +14453324242342421
             Zimbabwe (ZIM)    +34452334323242321
             Nigeria (NGA)     +34352162252332321
             Congo, De. Rep. (COD) +14463262242141321
             Sierra Leone (SLE) +25462233232232321
             Chad (TCD)        +14353162233232321
             Mozambique (MOZ)  +14373223232232322
             Angola (AGO)      +14364151331132321
             Yemen (YEM)       +44414161421232121
             Mauritania (MRT)  +45333221331122321
             Haiti (HTI)       +34152112341122211
                               |----------------
                               |APDEKGJLCHMBOIFQN
```

**Figure 3.** Guttman Scalogram.

## 5. Discussion

This study focuses on responding to one of the most complex problems of sustainability at the country level, and this is measuring compliance with the SDGs. Thus, this research represents the first attempt to measure tourism competitiveness in terms of sustainability by finding a relation between WEF indicators and SDGs. Namely, the WEF uses a quantitative methodology to measure destination competitiveness, while tourism indicators for SDGs are yet to be developed. Therefore, measuring the level of achievement of each SDG for a particular tourism destination is challenging because there is no consensus on quantitative indicators. This study investigated a correlation between all SDGs and appropriate WEF indicators and found that SDGs measure tourism competitiveness.

Secondly, it is worth highlighting the valuable contribution of the Rasch methodology used in this research. With its application, the model obtained allows analyzing the relationships that occur between the countries in the sample when studying them as a group and not individually. In this way, the results obtained for a specific country are contextualized when compared with the rest of the countries.

According to the World Tourism Organization (UNWTO), the procedure for measuring the SDGs is a long process, which requires the use of many economic resources to estimate the achievement of how the SDGs will be measured [14]. However, in this work, a model has been established that is applicable to multiple tourist destinations. Therefore, with this model, it will be possible to evaluate, until the procedure established by the UNWTO is resolved, the greater or lesser compliance with the SDGs by the countries. The Tourism Sustainable Development Index (TSDI) is the only proposal that allows measuring the level of sustainability of the tourism sector of countries from satellite data [14]. In this sense,

the calculation of this index shows some weaknesses that reveal its low statistical strength when establishing leadership positions for developing countries.

The relevance of the variables closest to the achievements of economic development and progress in the consideration of greater tourism competitiveness is evident in this research since the SDGs related to ending poverty are highly valued (SDG 1) [A], peace, justice, and strong institutions (SDG 16) [P], quality education (SDG 4) [D], gender equality (SDG 5) [E], and sustainable cities and communities (SDG 11) [K] as generators of highly competitive advantage to tourist destinations. This reveals that to increase the competitiveness of tourist destinations, public policies must be aimed at strengthening basic aspects of the well-being of the population. On the one hand, there must be a firm commitment from the administrations to the development of quality educational and health systems to which the population can have free and equal access. In this way, the countries will have an increasingly healthy and educated population, which may contribute to reducing poverty levels. For their part, tourism managers will be able to have workers trained in tourism, from which they can develop tourism services and activities that generate wealth and economic growth, resulting in the destination's competitiveness.

The scarce relevance is given to SDG 14 [N] (Life below water) as an indicator of the competitiveness of tourist destinations. This reflects that, to a large extent, the sea, the oceans, marine resources, and underwater life are not valued as a relevant part of the territory of the tourist destination on which action must be taken to improve sustainability. Although these data may be affected by the low score given to this item by non-coastal countries, the position it occupies at the end of the classification of relevant SDGs for tourism competitiveness reflects that marine, underwater, and coastal resources, as well as marine and coastal biodiversity, are not receiving sufficient performance by the countries. Therefore, the appropriate authorities in each country should recognize a better consumer acceptance and appreciation of the tourist destination directly related to its seabed and its coastline, with its attractions in water sports, beaches, and seascapes. That is why tourist destinations will have to increase their efforts to improve this SDG, with policies aimed at, for example, obtaining blue flags, carrying out campaigns against marine pollution or coastal protection actions, as well as marine habitats and species, in addition to improving the planning of the maritime space with more protected maritime spaces, limiting the effects of human actions at sea.

At a global level, it is worth highlighting the limited contribution of SDG 17 [Q] (Partnerships for the goals) to tourism competitiveness and sustainability. This could reflect the scarcity of effective international collaborations and agreements to make this planet more sustainable, suggesting an increase in efforts in this regard. These alliances would be especially useful for progress in developing countries.

Despite the implications, this study is not without its limitations. First, it should be clarified that, of a total of 195 countries recognized by the UN, 140 countries were analyzed, which are those that make up the sample analyzed by the WEF in 2019. For 55 countries, the WEF does not have data, and therefore, their tourism competitiveness in terms of sustainability could not be examined in this research. Second, all SDGs were included in the study. However, not all SDGs' targets could have been covered by the WEF's list of indicators. Only those indicators in which an appropriate correlation was found were included in the study. In addition, some SDGs are measured only by one WEF indicator, although this circumstance did not affect the statistical validity of the model, according to the analyses carried out.

The model obtained in this work, although useful, can be improved and constitutes an objective for future work. In addition, results obtained in the Guttman Scalogram (Figure 3) reveal that there are certain SGDs that, regardless of the level of socioeconomic development in the country, their degree of compliance is similar. This shows that more variables are involved in achieving the SDGs, although the level of development is one of the most important. In this way, because the response patterns of the countries could be analyzed and grouped according to different variables of discrimination, another future

line of research could be related to identifying the key variables that serve as a catalyst for the implementation of the SDGs.

**6. Conclusions**

A measurement model in this study has been proposed using data from the WEF. The proposed model made it possible to conduct, in an unprecedented way, a robust and statistically reliable measurement of the SDGs. In this way, the problem of measuring sustainability at the country level in accordance with the SDGs was overcome.

A classification of tourist destinations was obtained in relation to the competitiveness derived from sustainability. As expected, the most competitive countries in terms of sustainability are those with the highest level of socioeconomic development. Numerous European countries stand out, along with the United States. This can be related to the fact that developed countries are economically more powerful and, therefore, invest more in sustainable development policies. In contrast, the least developed countries and the developing countries clearly show their low level of achievement in compliance with the SDGs and, thus, a low level of tourism competitiveness derived from sustainability. In fact, this is a global problem that the UN is trying to tackle. To this end, the UN favors international support and alliances between countries with different levels of socioeconomic development with the aim of implementing the SDGs. However, it seems that this measure still has not achieved the desired results. This can be seen in the findings obtained in this study, confirming that SDG 17 [Q] (Partnerships for the goals) is one of the SDGs with the least influence on tourism competitiveness. In contrast, the most relevant SDGs for tourism competitiveness are related to prosperity and social guarantees.

**Author Contributions:** Conceptualization, I.T., Z.A.-R. and E.P.-L.; methodology, Z.A.-R. and V.T.D.-P.; software, V.T.D.-P.; investigation: I.T., Z.A.-R. and V.T.D.-P.; data curation: I.T., Z.A.-R., E.P.-L. and V.T.D.-P.; writing—original: I.T., Z.A.-R. and V.T.D.-P.; writing—review, editing and supervision, E.P.-L. and V.T.D.-P. All authors have read and agreed to the published version of the manuscript.

**Funding:** This research received no external funding.

**Conflicts of Interest:** The authors declare no conflict of interest.

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
