# Peer review of "Tourism Competitiveness versus Sustainability: Impact on the World Economic Forum Model Using the Rasch Methodology"

_sustainability, doi:10.3390/su151813700_

Round 1

Reviewer 1 Report

The aim of the paper is clear and is stated what the study found.

The title is informative and relevant and the references are relevant, recent and referenced correctly. However, when discussing about tourism competitiveness and it being directly related to the destination's ability to attract and satisfy tourists, I recommend that authors expand their references beyond sources 3 and 4. While Crouch & Ritchie's study (Crouch, G. I., & Ritchie, J. B. (1999). Tourism, competitiveness, and societal prosperity. Journal of business research, 44(3), 137-152.) stands as one of my suggestions, there exist numerous other pertinent sources worth considering.

The research methodology, particularly the section 3.1 Relations between WEF and SDGs, is excellently written and explained. This part of the article is a significant strength. However, Figure 1 needs to be redesigned to improve clarity as its current form is not readable.

I do not understand the purpose of including reference 38 on the reference list and on the line 334.

The statements in lines 553-555 need to be revised and further clarified. Personally, I find it challenging to conceive whether managers of tourist destinations can exert significant influence on SDG1 and SDG16. Even in the case of SDG4 and SDG5, their impact seems to be confined to the realm of education quality and gender equality, restricted only to the DMOs under their management.

On the lines 588-589 Figure 2 mention have to be replaced with Figure 3.

Although minor modifications may be necessary, the article's overall worth and its potential contribution to the academic literature on the connection between SDGs and tourism competitiveness are evident.

Author Response

First of all, thank you for your comments. They serve us to improve. All your comments have been solved and you can see them in the attachment.

Reviewer 2 Report

The submitted manuscript analyzes data regarding the issue of tourism competitiveness versus sustainability by means of the Rasch model. The paper could be improved in some aspects.
Major comments:
1.    293: Please set the formula in the text. Follow the instructions in the MDPI templates.
2.    293: Note that the formula refers to the rating scale model. The Rasch model has been originally proposed for binary item responses. Please clearly distinguish these terms in the manuscript.
3.    294ff.: Be careful of the proper use of the italic and non-italic font in mathematical symbols.
4.    Section 3: Please better motivate why one should use the Rasch model. In particular, why should the data fits the model and not the other way around? It seems entirely unclear to me why all countries and items should equally well fit the data. In contrast, analyzing the heterogeneous functioning of items with a factor analysis or a cluster analysis would be much more convincing to me.
5.    326: I would expect that this section has the title “Results.”
6.    336: It is incorrect to say that infit and outfit are “internal” and “external” statistics, respectively. You will not find a (serious) reference that states this.
7.    Table 2, Table 3: Please do not use a screenshot for the table. Arrange proper table formatting within the file.
8.    Sect. 4.2: Authors elaborate on Cronbach’s alpha. It is unclear to me why they discuss this and use the reliability stemming from the Rasch model. Reliability in the Rasch model is defined differently.
Minor comments:
9.    42 and others: I do not see why “17” is additionally included in parenthesis in this sentence. I do not see a reason for doing so.
10.    112 and others: In sentences, “&” must not be used in references. Write “Kunst and Ivandic” instead “Kunst & Ivandic.”
11.    174: typo “y 3.06”.
12.    249: typo “3.2-“

Author Response

First of all, thank you for your comments. They serve us to improve. Your comments have been solved and you can see them in the attachment. However, I would like to clarify some doubts that you have specified in your comments:

-
Section 3: Please better motivate why one should use the Rasch model. In particular, why should the data fits the model and not the other way around? It seems entirely unclear to me why all countries and items should equally well fit the data. In contrast, analyzing the heterogeneous functioning of items with a factor analysis or a cluster analysis would be much more convincing to me.
This is the characteristic that defines the Rasch model and differentiates it from other techniques, such as structural equations, for example. The data is adjusted to an ideal mathematical model and, based on the analysis of the mismatches, it is possible to identify which subjects and items do not represent the latent variable to be studied. This "generic" fit does not assume that the both data, subjects and items, fit the ideal model in a equally way. To evaluate the adjustment level (or validity), the adjustment statistics "INFIT" and "OUTFIT" are considered, determining that the adjustment level is adequate when the levels of these statistics are within the established intervals.
- It is incorrect to say that infit and outfit are “internal” and “external” statistics, respectively. You will not find a (serious) reference that states this.
In the chapter 7 of the book "Wright, B. D., & Stone, M. (1999). Measurement essentials", that you can see online (https://www.rasch.org/measess/me-all.pdf), the adjustment statistics, are defined in this way (page 53):

"These fit statistics are called "outfits" because they are heavily influenced by outlying, off-target,
unexpected responses. A useful alternative is to weigh residuals by the information they contain so that
the fit statistics are information weighted or "infits" and hence focus on inlying, on-target, unexpected
responses"
In relation with that, we have assumed that "inlying" and "outlying" are "internal" and "external" desadjusment factors

-
Sect. 4.2: Authors elaborate on Cronbach’s alpha. It is unclear to me why they discuss this and use the reliability stemming from the Rasch model. Reliability in the Rasch model is defined differently.

The Cronbach´s alpha analysis is elaborated and proportionated by Winsteps. In the Rasch model, the study of reliability Rasch model is carried out  through Cronbach´s Alpha in combination with specific indicators (reliability).

Reviewer 3 Report

After reading the manuscript, I can make the following recommendations and comments:

- in my opinion, the authors' idea is of interest to both a specialized and a wider audience; I share their stated goals to be useful useful to managers in tourist destinations;

- the topic is current and important for the strategic focus for the development of tourist destinations and enterprises;

- the study is well constructed and structured.

I would like to make the following recommendations:

1. regarding the Abstract

- it would be good to specify the methodology chosen to be used, as well as to avoid enumeration when presenting the results.

2. regarding the Introduction

- I would suggest that the authors consider editing the introductory interrogative sentence - line 32 of the Introduction. In this case, we have a scientific article, which should observe a certain character, and a similar question referring to the definition of the notion of sustainability could be transformed in a more appropriate way.

- following the logic of the text, in my opinion, it would be good to add a definition of competitiveness in general and in particular when referring to tourism. (Thus, the relationship and interdependence that the authors imply and that they try to explain and analyze would be better outlined.)

- another aspect that seems to me to be missing concerns the existing indices for measuring sustainability in tourism, where different indicators are also used, including at the level of a tourist destination - a country (such as - The Sustainable Tourism Index, and others). I understand that this goes beyond the focus of the particular study, but it would be good to mention them because they exist and are used.

3. regarding the Methodology

- I would only note that, in my opinion, it would be good to add why the Rasch model is suitable for the specific study - between lines 280-282.

4. regarding the Discussion

- I find that the results are well presented and illustrated in the Discussion section. Of course, the question remains whether it would not be better to have two separate independent sections - Results and Discussion. This comment is directly related to the next one I would like to make - I believe that it would be better as recommendations formulated on the basis of them to move points 5.1 and 5.2 to the end of the Discussion section. Thus, the overall balance of the article would be more definitive.

Author Response

First of all, thanks for your comments. They help us to improve. Your comments have been resolved and you can see them in the attached document.

Round 2

Reviewer 2 Report

The revised manuscript analyzes data regarding tourism competitiveness versus sustainability by means of the Rasch model. A few things remain to be addressed:
1.    Formula of the partial credit model, p. 8: “ni” must appear as subscripts of “X.” The symbol \gamma is not explicitly defined. The “interviewed country” has the symbol “v”, but it is “n” in the formula. The explanation of “x” does not make sense, in my opinion. Why should it be a latent variable?
2.    My sixth comment was unsatisfactorily addressed, line 356: I asked you to exclude the definition of internal and external statistics for infit and outfit. Your response made it worse because it demonstrated that your response is unrelated to the definition of the fit statistic. Just describe these statistics as mean squares fit statistics without referring to the metaphoric statement of “internal” and “external” location of the data. The unprofessional writing of the partial credit model formula seems to demonstrate that you do not entirely understand what you are writing. If the authors are criticized by a reviewer, they should first find fault with themselves.
3.    My third comment was unsatisfactorily addressed: I still do not see why a Rasch model should be ideal in measurement.
4.    My eighth comment was unsatisfactorily addressed. I still insist that you report the MLE reliability based on the partial credit model. The fact that Cronbach’s alpha is advocated in some Dinosaur Rasch modeling software is no argument that it is an inadequate reliability measure when using logit scores from Rasch modeling.

Author Response

Thank you very much for your comments about errors in the formula. We have revised and we have rewrite the formula of the Rasch Rating Scale. You can see that in the attached document (page 8). In that formula "x" is the latent variable because it allows to determine the level of latent variable in a single dimension by the probability of locating the latent variable in the linear continuum.

In regard about technique chosen in this work, the justification for the application of the Rasch model for the measurement resides in its usefulness to statistically validate a measurement instrument that uses ordinal scales, such as the model proposed in this work for the measurement of the Sustainable Development Goals (SDGs). The Rasch model transforms the raw data (ordinal variables) into logit measures (interval variables) on which to apply the statistical indicators.

We are very grateful for your comments regarding the fit statistics. We have modified its definition and avoided any mention of the terms "internal" and "external".

Thank you very much for your comments about the use of Cronbach's Alpha. In this sense, it is considered that Cronbach's Alpha is adequate to measure the reliability of the measurement instrument in Social Sciences since it is equivalent to the internal consistency coefficients when the items used for measurement admit multiple answers (polytomous items). In the Rasch model, particularly in the Rating Scale Model, Cronbach's Alpha refers to the separation between the measures of the country and item parameters, and reports the precision achieved with the proposed measurement. In this case, Cronbach's Alpha in combination with the specific "reliability" indicators, make it possible to identify whether the measures provided by the Rasch model are statistically reliable and accurate to measure sustainability in terms of tourism competitiveness, and thus confirm the main objective of this work.

Round 3

Reviewer 2 Report

---